# Neuropsychological Diagnosis and Assessment of Alexia: A Mixed-Methods Study

**DOI:** 10.3390/brainsci14070636

**Published:** 2024-06-25

**Authors:** Ahmed Alduais, Hessah Saad Alarifi, Hind Alfadda

**Affiliations:** 1Department of Human Sciences (Psychology), University of Verona, 37129 Verona, Italy; 2Department of Educational Administration, College of Education, King Saud University, Riyadh 11362, Saudi Arabia; 3Department of Curriculum and Instruction, College of Education, King Saud University, Riyadh 11362, Saudi Arabia; halfadda@ksu.edu.sa

**Keywords:** alexia, brain injury, neuropsychological assessment, neuropsychological diagnosis, reading impairment

## Abstract

The neuropsychological diagnosis and assessment of alexia remain formidable due to its multifaceted presentations and the intricate neural underpinnings involved. The current study employed a mixed-method design, incorporating cluster and thematic analyses, to illuminate the complexities of alexia assessment. We used the Web of Science and Scopus to retrieve articles spanning from 1985 to February 2024. Our selection was based on identified keywords in relation to the assessment and diagnosis of alexia. The analysis of 449 articles using CiteSpace (Version 6.3.R1) and VOSviewer (Version 1.6.19) software identified ten key clusters such as ‘pure alexia’ and ‘posterior cortical atrophy’, highlighting the breadth of research within this field. The thematic analysis of the most cited and recent studies led to eight essential categories. These categories were synthesized into a conceptual model that illustrates the interaction between neural, cognitive, and diagnostic aspects, in accordance with the International Classification of Functioning, Disability, and Health (ICFDH) framework. This model emphasizes the need for comprehensive diagnostic approaches extending beyond traditional reading assessments to include specific tasks like character identification, broader visual processing, and numerical tasks. Future diagnostic models should incorporate a diverse array of alexia types and support the creation of advanced assessment tools, ultimately improving clinical practice and research.

## 1. Introduction

Alexia, an acquired reading disorder, manifests in different forms, each delineated by distinct cognitive and neurological underpinnings [1,2]. In the synthesis of the existing literature, alexia is portrayed as a disorder that can occur as an isolated symptom or as part of an aphasia syndrome, stemming from disruptions to the linguistic, visual, or perceptual processes essential to reading [1,2]. Alexia can manifest in several forms, each with distinct characteristics and neurological underpinnings. Pure alexia involves the loss of reading ability while preserving writing and spoken word recognition. Surface alexia is characterized by difficulty reading irregularly spelled words, while phonological alexia involves struggles with reading unfamiliar or nonsense words. Deep alexia is marked by semantic reading errors, and hemianopic alexia is associated with reading difficulties due to visual field defects. Lastly, attentional alexia involves issues with grouping letters correctly to form words, related to deficits in visual attention and spatial processing [1,2,3,4,5].

The diverse etiologies of alexia, ranging from posterior cortical atrophy to acute disseminated encephalomyelitis, underscore the disorder’s complexity and the necessity for precise neuropsychological assessment to guide effective rehabilitation [1,3]. Recovery from alexia involves the re-engagement of alternative neural pathways, revealing the brain’s ability to compensate for disrupted reading processes [5]. Such findings advocate for a tailored approach to treatment, where interventions are directed by the specific nature and severity of the alexia presented. Collectively, these studies furnish a rich tableau of alexia, inviting continued exploration into the intricate interplay between brain, language, and cognition that defines this reading disorder.

The neural mechanisms of pure alexia have been illuminated through a confluence of neuropsychological studies that collectively underscore the complexity of this condition [5]. Pure alexia arises predominantly due to lesions in the left occipito-temporal lobe, particularly affecting the Visual Word Form Area (VWFA), which is pivotal for the rapid, parallel processing of words [5]. Recovery has been observed through the recruitment of alternative neural pathways, such as the spared occipital cortex and the right hemisphere, though these routes do not restore reading efficiency to pre-lesion levels, underscoring the critical role of the VWFA [5]. Further, disconnection alexia (alexia without agraphia, caused by a disconnection between the VWFA and language areas due to damage to the left occipitotemporal white matter) and cortical alexia (alexia with agraphia, caused by damage to the left angular gyrus resulting in impaired visual and linguistic processing) are two distinct subtypes of alexia distinguished by their lesion location and associated deficits [6]. Furthermore, researchers suggest that a general reduction in visual speed and span, rather than a specific orthographic deficit, may underpin the disorder, highlighting the impact on word recognition [7]. This is complemented by the findings of other studies arguing against reduced spatial frequency sensitivity as a general explanation for pure alexia, suggesting deficits at a higher level of visual processing [8]. Another team of researchers provides evidence of the diversity of etiological factors, from nonconvulsive status epilepticus to white matter damage, that can precipitate pure alexia, further complicating the neural underpinnings of the disorder [9,10]. Together, these studies [5,6,7,8,9,10] provide a comprehensive understanding of the multifaceted neural mechanisms of pure alexia, spanning from focal lesions to integrative visual processing deficits.

A thorough diagnostic process typically involves an assessment of anatomical correlates, which is complemented by neuropsychological assessments that among others discern phonological alexia, where the disruption is localized at the phonological processing stage, sparing oral expression and comprehension [3,10]. Additionally, the role of visual field defects in reading impairment is scrutinized, as seen in studies comparing eye movement patterns and fixation frequency during reading tasks (using eye movement patterns and fixation frequency), which differentiate pure alexia from hemianopic dyslexia and emphasize the importance of assessing visuo-motor coordination [4]. Diagnosis also involves examining the functional repercussions of alexia on language and other cognitive domains, such as memory and object recognition, to exclude related deficits [2,11].

Advances in neuroimaging techniques have significantly enriched the diagnostic process, enabling the longitudinal study of brain mechanisms and the recovery trajectory in pure alexia. Such studies have highlighted the compensatory activation of alternative neural pathways, despite the persistence of a slower, letter-by-letter reading pattern, and have underscored the critical role of the VWFA [5]. In some cases, alexia may manifest without agraphia, indicating a neurologically dissociated syndrome, which requires a nuanced approach to diagnosis, encompassing both behavioral and neurological assessments [1,12]. Keulen’s work on the theoretic basis underlying alexia further underscores the importance of differentiating between central and peripheral alexias and the utilization of eye movement neuroanatomy in diagnosis, thereby illuminating the multifactorial nature of the disorder and the need for a comprehensive diagnostic framework that integrates both cerebral pathology and cognitive function [13].

## 2. Purpose of the Present Study

The aim of this mixed-method design study is to rigorously explore the multifaceted domain of neuropsychological diagnosis and assessment of alexia, employing both quantitative cluster analysis and qualitative thematic analysis to yield a comprehensive understanding of the landscape. The scope of this study is to meticulously map the contours of current and foundational research, extracting salient themes and quantifiable patterns that emerge from the corpus of alexia literature. By synthesizing the insights gleaned from the top-cited works and the vanguard of recent studies, this research attempts to offer a nuanced portrayal of the advances and enduring questions within the field.

In the crucible of this academic inquiry, a conceptual model will be presented, designed to encapsulate the diagnosis and assessment paradigms that underpin alexia. This model will serve as a theoretical scaffold, integrating the diverse strands of alexic research—from neural substrates and cognitive processes to cultural and linguistic considerations—into a coherent framework. The interplay between empirical findings and theoretical postulations within this model aims to advance the discourse on alexia, offering a refined lens through which practitioners and researchers can navigate the complexities of this reading disorder. Through this study, we seek to provide a definitive reference point that will inform both clinical practice and future scholarly endeavors in the nuanced assessment and diagnosis of alexia, aligning with the International Classification of Functioning, Disability, and Health (ICFDH) framework.

## 3. Methods

### 3.1. Sampling

The quantitative phase of this study commenced with an analysis of 449 full articles written in English, published between 1985 and 2024 (February), retrieved for cluster analysis. These articles were sourced from the Web of Science and Scopus databases, encompassing a comprehensive selection of the literature on the neuropsychological diagnosis and assessment of alexia. For the qualitative phase, the thematic analysis was conducted using a curated selection of the top 25 cited documents and the 10 most-recent documents pertinent to alexia.

Search Terms (as of 23 February 2024):

‘Alexia assessment’ (Topic) or ‘Alexia evaluation’ (Topic) or ‘Alexia diagnosis’ (Topic) or ‘Alexia manifestations’ (Topic) or ‘Alexia disorder’ (Title) or ‘Alexia’ (Title)|Total Articles: 449.

### 3.2. Design

This mixed-methods study employed both quantitative and qualitative designs, with cluster analysis elucidating the existing data clusters on alexia and quantifying the extent of research in this field. This was followed by an in-depth thematic analysis, which aimed to extract and synthesize core themes from the identified literature.

### 3.3. Measures

The methodological rigor of this study was upheld by the deployment of sophisticated bibliometric and content analysis tools. For the quantitative phase, CiteSpace and VOSviewer software were harnessed to perform a cluster analysis—a statistical method that groups articles based on the similarity of their bibliometric features, such as keywords, citations, and co-authorship networks. These tools not only identified prevalent themes and trends within the corpus of literature but also facilitated the visualization of intricate relationships between studies, highlighting the most influential works and emergent topics within the field of alexia research.

For the qualitative phase, thematic analysis was meticulously conducted, grounded in the data illuminated by the cluster analysis. This involved a systematic examination of the top 25 cited studies alongside the 10 most-recent studies, with the aim of extracting recurrent and salient themes that characterize the neuropsychological diagnosis and assessment of alexia. The thematic analysis was pivotal in discerning the nuanced intricacies of the field, parsing out the subtle threads of continuity and divergence that weave through the literature.

The synthesis of these two analytical approaches culminated in the conceptual model, a theoretical construct devised to articulate a comprehensive framework for the neuropsychological diagnosis and assessment of alexia. This model was constructed based on the main findings from both the quantitative and qualitative analyses, encapsulating the depth and breadth of understanding garnered from the extensive review of the literature. The conceptual model serves as both a summative representation of the current state of knowledge and a heuristic guide for future research in the domain of alexia.

### 3.4. Procedure

The methodological approach involved the following procedural steps:Conducting a comprehensive search on Web of Science and Scopus databases using the specified search terms to collate data relevant to the neuropsychological diagnosis and assessment of alexia.Utilising CiteSpace and VOSviewer software to perform cluster analysis on the retrieved data.Selecting and retrieving the top 25 cited studies for thematic analysis to gain insights from seminal works in the field. To mitigate citation bias, we also included the 10 most-recent studies on the topic.Reviewing an additional 29 studies, selected based on their relevance to the topic, for inclusion in the introduction and discussion sections.Initiating thematic analysis by identifying themes derived from the cluster analysis results.Extracting data from the 35 studies and reviewing them with the research team for topic relevance.Presenting the extracted data in tables within the findings section.Developing a conceptual model for the neuropsychological diagnosis and assessment of alexia, utilizing key insights from both the cluster analysis and thematic analysis.

## 4. Findings

### 4.1. Cluster Analysis

Figure 1 is a visualization of the co-occurrence of keywords related to alexia, generated using VOSviewer software. The four clusters identified in the analysis are represented by different colors: red, green, yellow, and blue. The size of the circles in the figure represents the frequency of the keywords, and the lines connecting the circles indicate the co-occurrence of these terms in the analyzed texts. Overall, this figure provides a valuable overview of the different aspects and related concepts associated with alexia research. It highlights the distinct characteristics of pure alexia, its connection to aphasia, its differentiation from dyslexia, and other relevant topics in the field.

Figure 2 shows that the top-ranked item by bursts is optic aphasia, with bursts of 5.07. The second one is word form area, with bursts of 4.71. The third is rehabilitation, with bursts of 4.30. The fourth is word, with bursts of 4.24. The fifth is by letter reader, with bursts of 4.21. The sixth is interactive account, with bursts of 4.18. The seventh is hemianopia, with bursts of 3.60. The eighth is account, with bursts of 3.26. The ninth is lexical access, with bursts of 2.87. The tenth is recovery, with bursts of 2.76. The green line indicates the publication beginning and the end of each burst. On the other hand, the red line indicates the beginning of each burst and its end for each keyword.

Figure 3 and Table 1 systematically compartmentalize the expansive body of research into several distinct clusters, each representing a thematic aggregation of studies characterized by their relevance to particular facets of alexia (we only report the top 10 clusters). The ‘Label LSI’ (Latent Semantic Indexing) column encapsulates the core theme of each cluster, extracted through a computational technique that identifies patterns within text data, thereby distilling the essence of the literature’s focus. For instance, the recurring theme of ‘pure alexia’ across multiple clusters underscores the prevalence of this subtype in the research landscape.

The ‘Label LLR’ column is instrumental in pinpointing key phrases that are statistically significant within the context of each cluster, offering a deeper insight into the specific issues and concepts that are predominant within the scholarly discussions on alexia. These key phrases are derived from the log-likelihood ratio test, which assesses the probability that a given word’s frequency in a corpus is due to chance, thereby identifying terms most characteristic of the dataset. For instance, in Cluster 0, the key phrase ‘posterior cortical atrophy’ is significant due to its high LLR score, indicating that damage or atrophy in the posterior regions of the cortex is frequently discussed in relation to pure alexia. This reflects the clinical understanding that the posterior cortical regions, including the occipital lobe and the posterior parietal cortex, are crucial for the visual processing required for reading and often implicated in alexia when damaged or atrophied. Similarly, Cluster 1’s key phrase ‘visual field defect’ highlights the importance of visual spatial processing and the integrity of the visual field in reading ability. Visual field defects, such as hemianopia, can severely impact a person’s ability to read, and their presence is a significant consideration in the diagnosis and understanding of alexia.

In Cluster 2, ‘right-hemisphere form’ suggests that discussions within this cluster may revolve around the role of the right hemisphere in reading and how forms of alexia could be influenced by right-brain injuries. This can be particularly relevant for scripts that require spatial processing, a function often attributed to the right hemisphere. The key phrase ‘sequential processing’ in Cluster 3 alludes to the cognitive process of decoding sequences of letters or characters in reading. Difficulties with sequential processing could manifest as slower reading speeds or errors, which are characteristic symptoms of certain types of alexia.

Each of these key phrases, identified through their LLR scores, provides a thematic anchor around which the studies in a particular cluster revolve. They represent the nuanced and diverse aspects of alexia that are critical for neuropsychological assessment and diagnosis, from the anatomical and functional deficits to the cognitive processing challenges faced by individuals with this disorder.

### 4.2. Thematic Analysis

In an effort to distil the vast expanse of literature on the neuropsychological assessment of alexia, two pivotal tables have been constructed, synthesizing the seminal and contemporary contributions to the field. Table 2, encompassing the top 25 cited studies, serves as a testament to the foundational research that has shaped our current understanding of alexia. These landmark studies, through rigorous empirical inquiry, have elucidated a spectrum of neuropsychological facets—ranging from the neural correlates of alexia to the cognitive and linguistic underpinnings of reading disorders. They have collectively laid the groundwork for diagnostic criteria and therapeutic interventions, underscoring the complexity and heterogeneity inherent in alexic presentations. Table 3, delineating the 10 most-recent studies, captures the cutting-edge advancements that continue to refine and expand our knowledge base. This contemporary research, while building upon the insights of previous work, introduces novel perspectives to the assessment of alexia. It reflects a growing sophistication in methodological approaches, embracing cross-linguistic analyses and novel diagnostic technologies, thereby propelling the field toward a more nuanced and individualized approach to assessment. Together, these tables offer a compendium of intellectual milestones that both chronicle the historical trajectory of alexia research and plot a course for future explorations within this dynamic domain.

### 4.3. Categories of Neuropsychological Diagnosis and Assessment of Alexia

Table 4 provides a categorized summary of the key themes found across the top-cited and most-recent studies on alexia. Each category encapsulates a distinct aspect of alexia research, with examples of relevant studies that have contributed to the understanding of the neuropsychological assessment and diagnosis of alexia.

In the intricate field of neuropsychology, the assessment and diagnosis of alexia—a disorder characterized by the loss of the ability to read—require a comprehensive understanding of its diverse manifestations and underlying mechanisms. The neural substrates of reading have been a focal point in alexia research, with studies meticulously examining the specific brain regions and networks implicated in reading processes [37]. Other researchers contributed to this domain by elucidating disconnectivity fingerprints causally linked to dissociated forms of alexia, establishing a nuanced understanding of the fiber pathways that facilitate the visual and linguistic processes essential for accurate word reading [43]. Such insights are indispensable, not only for pinpointing lesion sites but also for tailoring rehabilitative strategies that address the specific neural disruptions experienced by individuals with alexia.

Cross-linguistic and cultural considerations, too, play a pivotal role in the neuropsychological landscape of alexia. Research in this arena has shed light on how alexia presents itself across different languages, taking into account the idiosyncrasies of various writing systems [39]. Another researcher further expanded this knowledge base by detailing how the dissociated reading and writing performance in Japanese kanji and kana scripts offer a window into the partially separable mechanisms undergirding these writing systems [45]. This body of work underscores the importance of culturally attuned diagnostic frameworks that accommodate the linguistic diversity of alexia presentations, ensuring that assessments are both accurate and culturally sensitive.

The realm of visual processing and perceptual deficits has also been extensively explored in alexia research. Studies have investigated the visual aspects of reading and the perceptual deficits that may underlie alexic disorders, such as crowding or hemianopic disturbances [41]. Other researchers highlighted the indispensable role of ophthalmological evaluation in recognizing urgent clinical conditions that extend beyond ophthalmic concerns [44]. This emphasis on the visual processing impairments integral to alexia emphasizes the need for comprehensive assessments that encompass visual fields and perceptual abilities, thereby ensuring a holistic approach to diagnosis and subsequent intervention. Collectively, these categories and the studies within them pave the way for a more informed and nuanced neuropsychological understanding of alexia, fostering advancements in both clinical practice and research.

### 4.4. A Conceptual Model for the Neuropsychological Diagnosis and Assessment of Alexia

The conceptual model for the neuropsychological diagnosis and assessment of alexia, in Figure 4, can be envisioned as a multi-layered framework, integrating various components that contribute to a comprehensive understanding of the disorder. This model is not linear but rather an interconnected system where each component can influence and inform the other. The following outlines the key elements of the model:

This foundational layer involves identifying the specific brain regions and networks associated with reading processes. Diagnosis begins with the localization of lesions or disconnectivity within the neural circuitry, such as the VWFA or language-dominant hemisphere, that may contribute to alexia [37,43]. The next layer acknowledges the importance of language-specific and cultural factors in the manifestation of alexia. The assessment should be sensitive to the script complexity and orthographic characteristics of the patient’s language, as these impact the nature of the reading disorder [39,45]. Concurrent with the above, evaluating visual processing abilities is critical. This includes assessing the presence of visual field defects, hemianopic disturbances, and abnormal visual crowding. Tools such as perimetry and detailed visual assessments are employed to delineate the scope and impact of visual deficits on reading [41,44]. Central to the model is the assessment of phonological and orthographic processing skills. These cognitive abilities underpin the decoding and encoding processes of reading and writing.

Neuropsychological tests that gauge phonological awareness, word repetition, and orthographic knowledge are utilized to diagnose specific subtypes of alexia [38,42]. This component considers the potential disconnection between crucial brain regions, rather than looking solely at focal damage. Diagnosis involves neuroimaging techniques such as diffusion tensor imaging to understand the disconnection patterns that may lead to various forms of alexia [23,46]. This layer involves assessing other cognitive and linguistic predictors that influence reading performance. Semantic abilities, working memory, and executive functions are evaluated to understand their contribution to the patient’s reading abilities and to inform targeted interventions [38,39]. This final layer focuses on the potential for recovery and the effectiveness of various rehabilitation strategies. Based on the findings from the previous layers, personalized rehabilitation plans are devised to improve reading abilities. This can include traditional language therapy, optokinetic therapy, and revascularization surgeries, depending on the etiology of alexia [36,46].

Throughout the entire model, the use of various clinical and diagnostic tools is emphasized. These range from traditional neuropsychological assessments to emerging digital platforms that can facilitate both the diagnosis and treatment of alexia [40,44]. In this conceptual model, each layer interacts with the others, creating a dynamic and responsive system for the diagnosis and assessment of alexia. The complexity of the disorder necessitates a multifaceted approach, where each component of the model provides a valuable perspective, contributing to a holistic understanding and tailored intervention strategies.

Our model in Figure 5 aligns with the framework of the ICFDH by addressing multiple dimensions of health and disability. The ICFDH emphasizes a comprehensive view of health that includes body functions and structures, activities, participation, and contextual factors such as environmental and personal factors. Our model integrates these components by assessing the anatomical and physiological aspects of alexia (body functions and structures), the impact on reading activities (activities), and the broader implications for academic and social participation (participation). Additionally, our model considers environmental factors like language and cultural context, as well as personal factors such as cognitive abilities and individual recovery potential. This alignment with the ICFDH framework ensures that our approach is holistic, addressing all facets of the patient’s experience and promoting personalized, person-centered care.

The conceptual model for pure alexia, aligned with the ICFDH framework, integrates multiple dimensions of health and disability. It encompasses body functions and structures, highlighting the neurological and visual mechanisms affected by a lesion in the left occipitotemporal region, specifically impacting the VWFA (as shown by an imaging technique (e.g., MRI results)), and including potential visual field defects like right homonymous hemianopia. Activities focus on the individual’s communication challenges, such as slow, letter-by-letter reading with significant difficulties in understanding written text and performing daily reading tasks, and involve therapeutic interventions like visual scanning exercises and the use of assistive technology, with progress monitored over time. Participation examines the impact on social interactions, noting potential isolation in social settings that require reading, adaptations in the professional environment such as transitioning to roles with reduced reading demands, and the overall quality of life, including self-reports of satisfaction with daily activities and family observations regarding mood and independence. Contextual factors include environmental factors, like support from family and access to speech therapy and assistive technologies, and personal factors such as the individual’s developmental and health history, motivation, resilience, and proactive behavior in seeking therapy. This comprehensive model ensures a holistic understanding and management plan for individuals with pure alexia.

## 5. Discussion

The present study aimed to dissect the intricacies of the neuropsychological diagnosis and assessment of alexia, utilizing both cluster and thematic analyses to shed light on the prevailing trends and seminal works in this domain. The cluster analysis elucidated the concentration of research efforts in specific areas, while the thematic analysis provided a rich narrative of the evolution and current state of alexia diagnosis. This dual approach revealed a multifaceted view of alexia, capturing both the breadth of the disorder and the depth of individual studies. In particular, the conceptual model used in this study served as a beacon, guiding the synthesis of findings and facilitating a deeper understanding of the nuanced interplay between neural substrates, cognitive impairments, and diagnostic strategies.

The examination of alexia’s various forms, etiologies, and neural mechanisms within the context of the previous literature highlights the complexity of the disorder and emphasizes the importance of a comprehensive diagnostic framework that integrates both cerebral pathology and cognitive function. For instance, in one study the authors provided foundational insights into the anatomical basis of pure alexia, identifying the lesion in the paraventricular white matter of the left occipital lobe as a key correlate [10]. This finding has been instrumental in shaping the diagnostic protocols that often commence with neuroimaging to pinpoint the locus of cerebral injury. Building on these anatomical insights, ref. [4] highlighted the specificity of visual field defects in pure alexia, advancing our understanding of the role of visuo-motor impairments. The manifestation of phonological alexia as detailed by [3] further refined the diagnostic spectrum, distinguishing it from other forms based on the affected cognitive stage.

As the discourse on alexia diagnosis advances, ref. [2,11] have emphasized the integration of psycholinguistic approaches into clinical assessment, acknowledging the complexity of alexia beyond mere neurological disruption. This evolution of thought is mirrored in the contemporary perspective where a neuroanatomical and eye movement-based approach to alexia diagnosis is advocated, as detailed by [13]. Such an approach underscores the necessity of a dynamic and comprehensive diagnostic framework that can accommodate the diverse presentations of alexia.

The synthesis of findings from this study, in conjunction with the established literature, bolsters the argument for a granular and multifactorial diagnostic approach. The use of the conceptual model in this study has provided a structured avenue to navigate the complexities of alexia, drawing from both the cluster and thematic analyses. This strategy has not only validated the historical underpinnings of alexia diagnosis but has also laid the groundwork for future research that may further refine and expand upon the current diagnostic methodologies. Through this methodological synthesis, the study has offered a contemporary perspective on an age-old disorder, proposing a diagnostic pathway that is both informed by historical knowledge and adaptable to future discoveries.

The complexity of alexia diagnosis is further exemplified by the interplay between peripheral and central deficits, as evidenced by the findings of [47], who challenged the notion of a singular phonological deficit in pure alexia. This underscores the importance of considering both peripheral visual processing and central linguistic processing in the diagnostic process. The study by [5], which documented recovery from pure alexia through alternative neural pathways, speaks to the dynamic nature of the brain’s compensatory mechanisms and the potential for neuroplasticity, a factor that should be meticulously considered in the prognosis and therapeutic planning for individuals with alexia. These insights necessitate a diagnostic approach that is not only responsive to the immediate presentation of alexia but also anticipatory of its longitudinal trajectory.

In the broader context of neurological conditions, the works of [1,12] exemplify the diagnostic challenges posed by the variability in etiology, such as multiple sclerosis and acute disseminated encephalomyelitis. These cases highlight that alexia can emerge as a symptom within a constellation of neurological disturbances, thereby requiring a multidisciplinary diagnostic strategy that can distinguish alexia from concomitant neurological manifestations. The nuanced understanding of these etiological complexities, as gleaned from the thematic analysis, enriches the conceptual model used in this study, emphasizing the need for a tailored diagnostic approach that aligns with the individual’s unique clinical presentation.

In synthesizing these findings within the context of the previous literature, the present study reaffirms the necessity for an expansive diagnostic lens, sensitive to both the cerebral and cognitive dimensions of alexia. The conceptual model derived from this study offers a structured framework for navigating these dimensions, integrating findings across anatomical, behavioral, and cognitive domains. This model propels the field towards a more holistic understanding of alexia, fostering a diagnostic paradigm that is as multifaceted as the disorder itself. Consequently, the model not only encapsulates the current state of knowledge but also serves as a scaffold for future research, potentially guiding the development of innovative diagnostic tools and targeted interventions that reflect the complexities of alexia.

## 6. Limitations

This study’s design, while comprehensive, is not without limitations. The reliance on the published literature sourced from Web of Science and Scopus may inadvertently exclude relevant grey literature or studies published in languages other than English, thus potentially biasing the findings towards research that has been more widely disseminated and recognized within academic circles. Furthermore, the retrospective nature of cluster and thematic analyses, although insightful, may not capture the real-time evolution of diagnostic practices or account for unpublished expert knowledge and clinical experiences that often guide assessment strategies. Additionally, the intricate process of constructing a conceptual model, while methodologically sound, may oversimplify the multifarious nature of alexia, potentially underrepresenting less-common or emerging subtypes of the disorder. These considerations highlight the importance of ongoing research and the continuous refinement of diagnostic models to accommodate the diversity and complexity inherent in alexia.

## 7. Implications

The implications of this study are multifaceted, extending across clinical, research, and theoretical domains. Clinically, the findings underscore the importance of adopting a heterogeneous diagnostic approach to alexia, one that is attuned to the individual nuances of each patient’s condition and underpinned by a robust understanding of the disorder’s neural and cognitive bases. For researchers, the study provides a conceptual framework that may inform future inquiries into the diagnosis and assessment of alexia, suggesting avenues for innovative methodological applications and interdisciplinary collaboration. Theoretically, the study contributes to the discourse on alexia by reinforcing the need for a dynamic and integrative model of diagnosis, bridging the gap between empirical evidence and clinical practice. The study’s insights have the potential to refine diagnostic criteria, enhance the efficacy of therapeutic interventions, and deepen the understanding of the neurocognitive underpinnings of alexia.

## 8. Conclusions

This study has systematically examined the landscape of alexia diagnosis and assessment, offering a rich synthesis of the disorder’s multifaceted nature through both cluster and thematic analyses. The resultant conceptual model encapsulates the intricate interplay between neural substrates, cognitive processes, and diagnostic strategies, providing a nuanced framework that aligns with both historical perspectives and contemporary insights. Despite its limitations, the study sets the stage for future explorations into alexia, challenging clinicians and researchers alike to consider the complex spectrum of alexia within their practices. As the field moves forward, it is imperative that the diagnostic models continue to evolve, informed by a growing body of research that reflects the diversity of alexia presentations and the individual stories behind each diagnosis.

## Figures and Tables

**Figure 1 brainsci-14-00636-f001:**
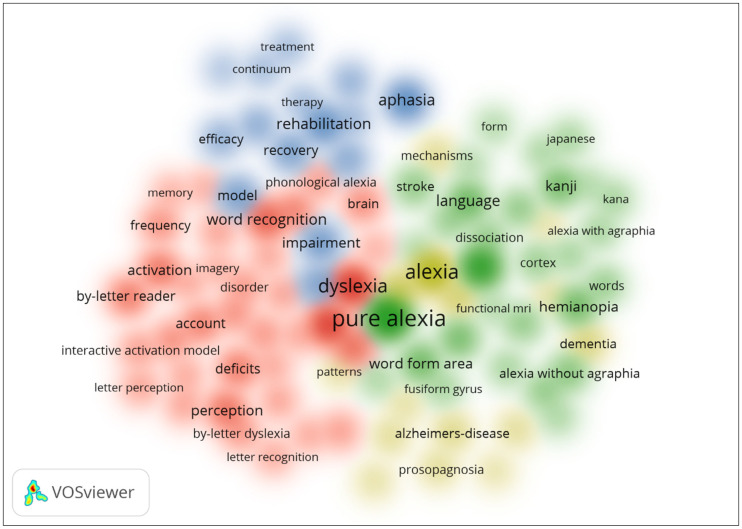
Density of visualization of word co-occurrence in alexia.

**Figure 2 brainsci-14-00636-f002:**
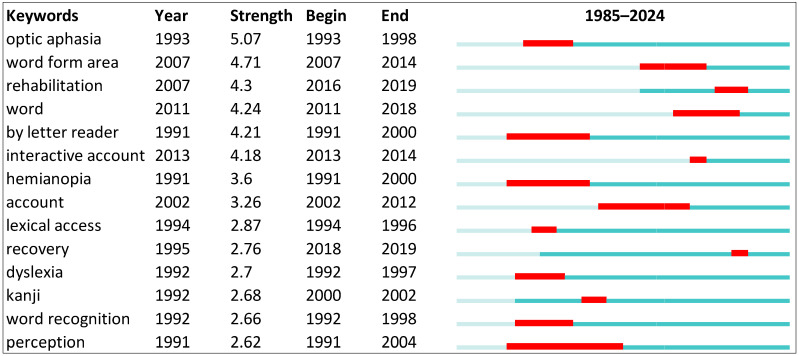
Top 10 keywords with the strongest citation bursts.

**Figure 3 brainsci-14-00636-f003:**
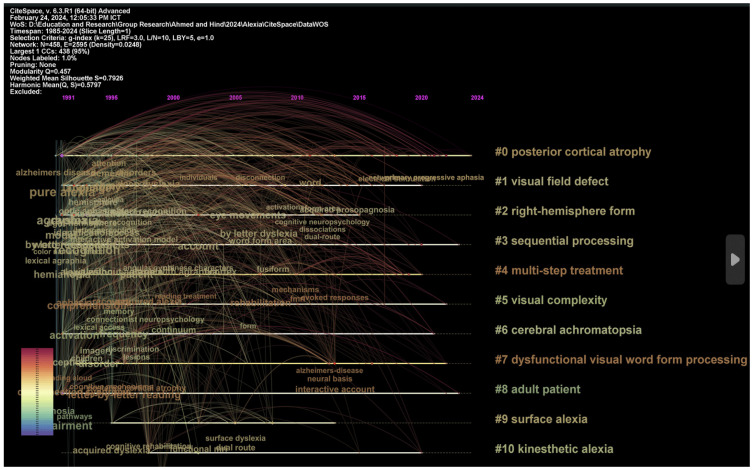
Visualization of top clusters in alexia.

**Figure 4 brainsci-14-00636-f004:**
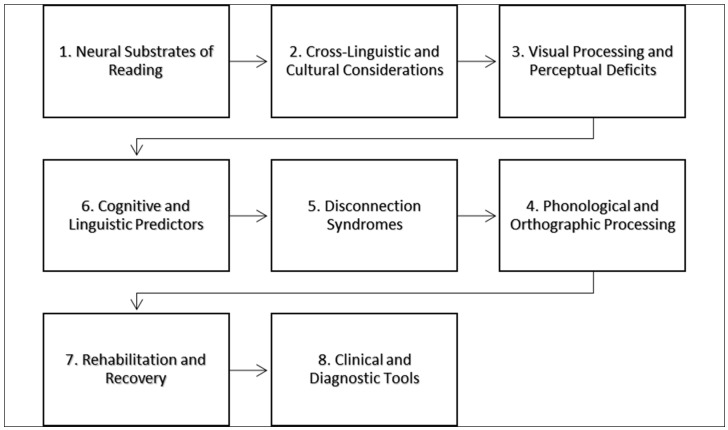
A conceptual model for the neuropsychological diagnosis and assessment of alexia.

**Figure 5 brainsci-14-00636-f005:**
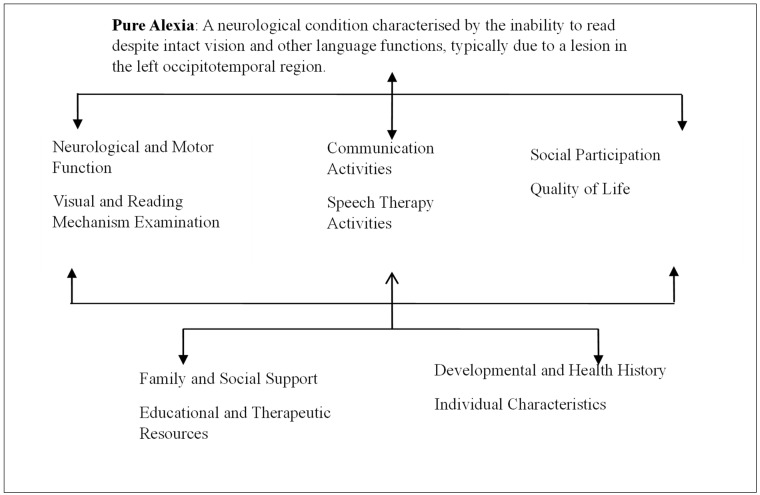
Diagnosis and assessment of pure alexia according to the ICFDH framework.

**Table 1 brainsci-14-00636-t001:** Summary of the 10 largest clusters.

Cluster ID	Size	Silhouette	Label (LSI)	Label (LLR)	Label (MI)	Average Year
0	70	0.762	pure alexia	posterior cortical atrophy (193.89)	alexia (1.73)	2009
1	64	0.778	pure alexia	visual field defect (64.28)	visual memories (1.41)	2005
2	52	0.665	pure alexia	right-hemisphere form (69.53)	alexia (1.01)	2001
3	47	0.833	pure alexia	sequential processing (74.86)	cortical network (0.44)	2006
4	40	0.775	multi-step treatment	multi-step treatment (123.64)	successful treatment (0.28)	2009
5	39	0.752	pure alexia	visual complexity (65.3)	by-radical reading strategy (0.23)	2002
6	33	0.878	pure alexia	cerebral achromatopsia (54.72)	pure alexia (0.14)	2002
7	26	0.754	pure alexia	dysfunctional visual word form processing (58.37)	posterior reversible encephalopathy syndrome (0.16)	2007
8	24	0.891	adult patient	adult patient (52.36)	pure alexia (0.16)	2004
9	21	0.913	following semantic mediation treatment	surface alexia (46.52)	pure alexia (0.17)	2004

LLR: Log-Likelihood Ratio, a statistical measure for term-cluster association strength. LSI: Latent Semantic Indexing, a technique for identifying underlying semantic themes in a document collection. MI: Mutual Information, a measure of mutual dependence between two variables, such as a term and a cluster. Average year refers to the mean publication year of articles within each cluster, indicating when research was most active in each topic.

**Table 2 brainsci-14-00636-t002:** A synthesis of the top 25 cited studies in the neuropsychological diagnosis and assessment of alexia.

No.	Citation	Aim	Findings	Takeaway from the Study about the Diagnosis and Assessment of Alexia
1	[14]	To investigate the identification processes of single characters in a pure alexic patient and the impairment causing the reading disorder.	It was found that the patient had a selective impairment in the identification of alphanumeric stimuli, attributed to deficits in selective processing mechanisms.	The study used a detailed analysis of single-character processing to assess alexia, implying a need for tasks that isolate character identification processes in diagnosis.
2	[15]	To challenge the notion that pure alexia is solely an orthographic processing disorder, by examining its relation to visual complexity in object identification.	Results indicated that pure alexic patients displayed impairments in visual object identification, suggesting a more general visual processing deficit rather than a purely orthographic one.	Alexia was assessed through object identification tasks, suggesting that diagnosis should consider broader visual processing abilities beyond just reading.
3	[16]	To examine whether word and face recognition deficits could co-occur, suggesting overlapping neural mechanisms.	The study found that prosopagnosic patients had word recognition deficits and pure alexic patients had face recognition deficits, indicating intertwined neural processes.	Assessment of alexia may benefit from including face recognition tasks, as deficits in word and face recognition may be related.
4	[17]	To explore the causes of the word length effect in pure alexia and determine whether it results from a spatial impairment or a letter activation deficit.	Findings suggested that pure alexia is a nonspatial disorder that impairs the rapid processing of single letters, challenging spatial impairment theories.	Diagnosis of alexia should consider the possibility of nonspatial visual disorders affecting letter activation, not just spatial impairments.
5	[18]	To investigate the dissociation between naming and comprehension abilities in the number processing domain of pure alexia patients.	The patient showed a preservation of some calculation abilities despite severe reading errors, highlighting task-dependent processing capabilities.	Assessment of alexia can involve numerical processing tasks to reveal preserved cognitive abilities that reading tasks might not uncover.
6	[19]	To explore the differential impact of task demands on number identification performance in patients with pure alexia.	Findings indicated variability in number identification performance based on task context, with preserved abilities in certain conditions despite severe reading impairments.	Task-dependent evaluation is crucial in alexia assessment, as patients may demonstrate residual abilities in numeracy that are not evident in reading tasks.
7	[20]	To investigate implicit reading abilities in patients who show explicit impairment in word recognition.	Patients performed above chance on lexical decision and semantic categorization tasks, despite inability to explicitly identify words, suggesting some preserved reading capabilities.	Implicit reading abilities in alexia can be assessed through lexical decision and semantic categorization tasks, revealing preserved cognitive functions.
8	[21]	To test the hypothesis that patients with pure alexia have access to distinct reading procedures based on task demands.	When tasked with naming words, the patient used a letter-by-letter strategy, whereas in lexical decision or semantic judgments, a whole-word strategy was employed, supporting the distinct procedures theory.	Different reading strategies may be employed by alexic patients, indicating the need for diverse task demands in diagnostic reading assessments.
9	[22]	To assess the co-occurrence of alexia and prosopagnosia and their potential association with object agnosia.	The patient displayed typical letter-by-letter reading and impaired face recognition but intact object recognition, challenging predictions that alexia and prosopagnosia necessarily co-occur with object agnosia.	Assessments of alexia should not presume associated object agnosia; instead, a nuanced approach considering different visual recognition abilities is required.
10	[23]	To understand the connectivity of the VWFA and its role in reading through the study of a pure alexic patient’s brain connectivity before and after surgery.	Progressive selective degeneration of the inferior longitudinal fasciculus (ILF) was observed, while the VWFA remained intact, highlighting the ILF’s critical role in normal reading and its disruption in pure alexia.	Diagnosis of pure alexia should include imaging methods like diffusion imaging to detect disconnection syndromes affecting reading capabilities.
11	[24]	To evaluate if the reading difficulty in pure alexia is due to a general visual impairment or a deficit specific to reading mechanisms.	The study supported the visual impairment hypothesis, showing that a pure alexic patient was impaired in specific visual tasks related to the hypothesized underlying deficiency.	Diagnosis of pure alexia should include a broader range of visual perceptual tasks to determine if a general visual impairment underlies the reading difficulty.
12	[25]	To delineate the neuroanatomical substrate of associative visual agnosia and its relationship with alexia and prosopagnosia.	The examination revealed that associative agnosia and alexia can occur without prosopagnosia and are associated with unilateral lesions in specific brain regions.	Alexia assessments should consider the possibility of associative visual agnosia and investigate the integrity of specific cortical regions and white matter tracts.
13	[26]	To describe a case of posterior cortical dementia presenting with alexia and to document corresponding neuroimaging findings.	MRI and PET scans revealed asymmetric bioccipitoparietal atrophy and hypometabolism, particularly on the left side, which correlated with the reading and other cognitive difficulties.	Neuroimaging should be integrated into the alexia assessment to identify potential atrophy and metabolic deficits in the posterior cortical regions.
14	[27]	To investigate the progression from deep alexia to phonological alexia and to conceptualize these conditions as points on a continuum.	Case studies showed a predictable pattern of symptom succession, suggesting a continuum of severity from deep alexia to phonological alexia.	Assessment of alexia should account for the continuum of severity and consider potential recovery patterns when diagnosing and planning treatment.
15	[28]	To provide evidence that deep alexia and phonological alexia share common deficits and neurological systems.	Two patients exhibited a transition from deep alexia to phonological alexia, supporting the notion of a shared continuum of deficits.	The continuum of alexic disorders should inform assessment strategies, recognizing that different types of alexia may transition over time.
16	[29]	To examine the role of the occipital cortex in Braille reading through the case of an early blind woman with bilateral occipital stroke.	The patient, a proficient Braille reader, lost this ability after the stroke, suggesting the involvement of the occipital cortex in Braille reading for the blind.	Diagnostic procedures for alexia in blind individuals should consider the potential impact of occipital cortex damage on Braille reading abilities.
17	[30]	To explore the structural anatomy of pure and hemianopic alexia via single-word reading in patients and controls using PET.	The study located the most posterior lateralized response involved in reading at the left occipitotemporal junction, with damage here linked to pure alexia.	Alexia assessments should distinguish between pure and hemianopic alexia, which may require different rehabilitative strategies.
18	[31]	To investigate differences in lesion sites between patients with hemianopic alexia and pure alexia to aid in diagnosis and rehabilitation.	Patients with pure alexia had additional lateral damage to the posterior fusiform gyrus, differentiating them from those with hemianopic alexia.	The examination of lesion sites can help differentiate types of alexia, which is crucial for directing appropriate rehabilitative techniques.
19	[32]	To identify the specific brain regions involved in reading and determine their role in alexia.	Stimulation of the dominant posterior fusiform and inferior temporal gyri induced alexia, suggesting their crucial role in reading.	Diagnostic procedures should consider the basal temporal region’s involvement in reading, potentially using brain stimulation techniques for assessment.
20	[33]	To investigate the case of a patient with a selective deficit in reading and naming visual objects, suggesting a failure to access graphemic representations.	The patient could not determine if letter pairs had the same name despite intact visual processing, indicating a disconnection between visual processing and graphemic representation.	Assessment of alexia should consider the integrity of the pathway from visual processing to graphemic representation access, going beyond surface-level reading difficulties.
21	[34]	To evaluate the reading performance of a Japanese patient with progressive aphasia and surface alexia, particularly with kanji and kana scripts.	The patient exhibited a pattern of alexia that was consistent with the interaction between word frequency and neighborhood-based consistency, differentially affecting kanji and kana reading.	Diagnosis of alexia in different writing systems should take into account language-specific factors, such as character frequency and consistency, to understand the nature of the reading deficit.
22	[4]	To determine the contribution of right-sided visual field defects to reading difficulties in pure alexia.	Eye movement patterns during reading differed between pure alexic and hemianopic dyslexic patients, suggesting that visuo-motor impairments in pure alexia are not solely due to visual field defects.	Diagnostic assessment of pure alexia should include detailed analysis of eye movement patterns during reading, not just visual field testing.
23	[35]	To investigate a case of verbal alexia and agraphia without musical alexia and agraphia in a blind organist, examining the independence of linguistic and musical competencies.	Despite profound verbal aphasia and alexia in braille, the patient retained musical abilities, suggesting distinct cognitive processing for language and music.	Diagnostic assessment of alexia should consider the potential dissociation between verbal and non-verbal (musical) symbolic systems, which may remain intact.
24	[36]	To test if optokinetic therapy can improve reading speeds in patients with hemianopic alexia by affecting reading saccades into the blind field.	The therapy improved reading speed and specifically affected rightward reading saccades, providing evidence for its effectiveness in hemianopic alexia.	Alexia therapy can include optokinetic therapy to specifically target and improve reading saccades, aiding in the rehabilitation of hemianopic alexia.
25	[7]	To determine if pure alexia is characterized by deficits in visual processing that are selective to words or letters, or if it reflects a broader visual impairment.	Pure alexia was associated with a general reduction in visual speed and span, not restricted to letters or words, affecting reading disproportionately.	Diagnosis of pure alexia should not only focus on word and letter recognition but also on broader aspects of visual processing, such as visual speed and span.

Note: Together, these studies provide a diverse understanding of alexia, revealing the need for a multifaceted approach to diagnosis and assessment that considers visual processing, specific brain region involvement, effects of rehabilitation techniques, and even the potential dissociation between different cognitive processes.

**Table 3 brainsci-14-00636-t003:** A synthesis of the 10 most-recent studies in the neuropsychological diagnosis and assessment of alexia.

No.	Citation	Aim	Findings	Takeaway from the Study about the Diagnosis and Assessment of Alexia
1	[37]	To document and explain a case of right hemi-alexia, potentially caused by intra-hemispheric disconnection affecting the VWFA.	The patient demonstrated a severe reading deficit in the right visual field, unrelated to object recognition, which resolved after tumor resection.	Right hemi-alexia can be diagnosed with careful perimetry and should consider the possibility of intra-hemispheric disconnection, highlighting the importance of lesion location and functional recovery post-intervention.
2	[38]	To examine common cognitive processes that support both spoken and written language, and modality-specific skills in individuals with language impairments.	Semantics and phonological skill were common predictors of language performance across modalities, with phonology showing the greatest deficit.	Assessment of alexia should not only focus on reading and writing but also consider underlying cognitive processes, such as phonological skills, that support language performance.
3	[39]	To analyze the clinical manifestations of alexias and agraphias in Spanish and the effect of demographic variables.	Reading and writing difficulties paralleled the severity of the oral language disturbances, with age and schooling impacting performance.	Diagnosis of alexia in Spanish-speaking patients should consider the parallel nature of oral and written language difficulties and be informed by demographic variables.
4	[40]	To argue for the importance of assessing written discourse in people with alexia and agraphia, especially in the context of digitalization.	No specific results described; the paper advocates for the assessment and treatment of written discourse abilities in PwAA.	Digital tools should be developed and utilized for the assessment and rehabilitation of written discourse abilities in patients with alexia and agraphia, to improve social and digital participation.
5	[41]	To explore perceptual classification deficits in pure alexia and prosopagnosia, challenging the notion of category-specific agnosias.	The study found complementary deficits in perceptual classification tasks, with a double dissociation between two classic cases in visual agnosia.	Diagnostic assessments of alexia should consider broader perceptual classification abilities and the potential cross-links with other visual agnosias, like prosopagnosia.
6	[42]	To describe a case of apraxia of speech, phonological alexia, and agraphia after resection of a brain tumor and explore their interrelations.	The patient showed isolated deficits in speech-motor planning and nonword reading and spelling, suggesting a shared process of motorphonological sequencing.	Diagnosis of alexia should consider the possible role of speech-motor planning regions and the interplay with reading and writing nonwords.
7	[43]	To identify fiber pathways related to different visual and linguistic processes needed for reading by examining neurosurgical patients with alexia.	Disconnectivity patterns of specific white matter tracts were associated with different subtypes of alexia, providing anatomical insights into cognitive models of reading.	Alexia assessments should include advanced imaging techniques to identify white matter disconnectivity patterns that may underlie different forms of alexia.
8	[44]	To present a case of pure alexia following posterior cerebral artery occlusion and emphasize the role of ophthalmological evaluation.	The patient’s reading difficulties and right superior homonymous quadrantanopia were associated with left posterior cerebral artery occlusion.	Diagnosis of pure alexia should be multidisciplinary, involving both neurological and ophthalmological evaluations to identify underlying vascular causes.
9	[45]	To analyze dissociated reading/writing performance in kanji and kana scripts in Japanese alexia and agraphia, and to understand the underlying mechanisms.	Lesion locations affected kanji and kana processing differently, with the left occipitotemporal area being important for both, but specific regions being associated with distinct symptoms.	Diagnostic assessments for alexia in Japanese should consider the script-specific pathways and the possibility of dissociated impairments in kanji and kana.
10	[46]	To investigate if revascularization can improve the reading and writing difficulties in kanji in patients with moyamoya disease.	Following surgical revascularization, patients with moyamoya disease showed dramatic improvement in their language dysfunction, specifically alexia with agraphia for kanji.	In cases of moyamoya disease, assessment of alexia should include monitoring of cerebral perfusion, and revascularization surgery should be considered as a potential treatment for language dysfunction.

Note: These recent studies highlight the complexity of alexia, showcasing the influence of neurological, linguistic, and cultural factors. They underscore the need for personalized, multifaceted diagnostic approaches and suggest potential new avenues for treatment and rehabilitation.

**Table 4 brainsci-14-00636-t004:** A categorization of the top 25 cited and 10 most-recent studies in the neuropsychological diagnosis and assessment of alexia.

No.	Category	Elaboration	How Studies Accounted for the Category	Sample Studies
1	Neural Substrates of Reading	Exploring the specific brain regions and networks involved in reading processes, including localization of functions within the brain.	Studies investigate the neural correlates of alexia, often using neuroimaging to identify damage or disconnectivity in language-processing areas.	[37,43]
2	Cross-Linguistic and Cultural Considerations	Addressing how alexia manifests in different languages and cultural contexts, considering the impact of writing systems on reading disorders.	Research reflects on the influence of language-specific factors, such as script complexity and orthography, on the neuropsychological assessment and diagnosis of alexia.	[39,45]
3	Visual Processing and Perceptual Deficits	Focusing on the visual aspects of reading and the perceptual deficits that may underlie alexic disorders, such as crowding or hemianopic disturbances.	Studies delve into the visual processing impairments in alexia, often highlighting the importance of assessing visual fields and perceptual abilities.	[41,44]
4	Phonological and Orthographic Processing	Investigating the deficits in phonological and orthographic processing that contribute to reading disorders.	Research emphasizes the role of phonological and orthographic processing skills in reading, influencing diagnosis and treatment approaches.	[38,42]
5	Disconnection Syndromes	Examining cases where alexia arises from the disconnection between critical brain regions, rather than focal damage to one area.	Studies reveal different forms of disconnectivity that lead to alexia, which is significant for understanding the underlying neuropathology.	[23,46]
6	Cognitive and Linguistic Predictors	Analysing cognitive and linguistic abilities that predict reading performance and their relevance to alexia.	Research identifies cognitive and linguistic predictors that are crucial for neuropsychological assessment, influencing diagnosis and prognosis.	[38,39]
7	Rehabilitation and Recovery	Discussing the potential for recovery and the effectiveness of various rehabilitation strategies in alexia.	Studies provide insights into the recovery process and the impact of different rehabilitation methods on the improvement of reading abilities.	[36,46]
8	Clinical and Diagnostic Tools	Addressing the tools and methods used for the clinical diagnosis and assessment of alexia, including digital tools and traditional neuropsychological tests.	Studies describe the use of various assessment tools, from digital platforms to established neuropsychological tests, to diagnose and quantify alexia.	[40,44]

## Data Availability

The data presented in this study are available on request from the first author.

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
