# Peer review of "Neuropsychological Diagnosis and Assessment of Alexia: A Mixed-Methods Study"

_brainsci, 2024, doi:10.3390/brainsci14070636_

Round 1

Reviewer 1 Report

Comments and Suggestions for Authors

Reviewer

Comments to the author:

Manuscript ID: brainsci-3051453

Title:Neuropsychological Diagnosis and Assessment of Alexia: 2 A Mixed-Methods Study”

Thank you for the opportunity to review this manuscript. The issue of the manuscript is interesting and has significant clinical value for clinicians who work in the diagnosis and assessment field of people with alexia.

Abstract

-The abstract is more than 200 words which is the upper limit the journal permits.

-The method subsection is missing. Authors should add the bibliographic base chosen to retrieve their articles and the chronological interval of retrieval.

-The results and conclusion subsection are too wordy. Be precise and clear presenting your results and conclusions. 

Introduction

Line 36. A citation is missing.

Lines 39-44. I suggest you refer to all subtypes of alexia and describe how the characterization of each type is made.

Line 54. Delete the phrase: “an acquired reading disorder”. The description of each type of alexia should be described in the first paragraph of your manuscript.

Line 56. Citations are missing.

Line 61. Form the citation according to the guidelines of the journal.

Lines 62-63. Explain what disconnection alexia and cortical alexia are and put the explanation into a parenthesis beside each term e.g. disconnection alexia (alexia without agraphia) and cortical alexia….

Line 70. Replace the word “Researchers” with “studies”

Lines 82-83. Delete these lines. In the above two paragraphs, the authors describe the anatomical regions involved in pure alexia. Instead write:A thorough diagnostic process typically involves an assessment of anatomical correlates, which is complemented by neuropsychological assessments that….”

Lines 86-88. You should refer to what clinicians use to assess visual-motor coordination and distinguish between pure alexia and hemianopic dyslexia.

Lines 94-96. Move these lines higher and put them at the end of the previous paragraph, after the tasks utilized to assess visual-motor coordination.

Overall, the whole introduction needs re-organization to be reader-friendly. At this point, the introduction lacks fluidity and connectivity. My suggestions to the authors are:

1.     Describe in the first paragraph the types of alexia and their characteristics.

2.     In the third and fourth paragraphs associate each subtype of alexia with the underlying neural mechanisms affected.

3.     In the next paragraph present the assessment battery for the diagnosis and differential diagnosis of alexia.

4.     Finally, describe why it is important for the treatment of alexia to differentiate between different subtypes and how neuroimaging techniques can help in this direction.

Methods

In sampling refer to the duration of the bibliographic search and if any criteria of exclusion for the articles included were used e.g. regarding the year of publishing.

Line 176. It is not clear to me how these 29 articles were detected. Please clarify.

Line 197-203. Please rephrase these lines to be easier to understand.  I would suggest deleting the cluster in which each keyword is included since the number of clusters that the cluster analysis provided has not been presented yet.

Line 209. In this line, you claim that the cluster analysis returned 10 clusters but in Figure 3 one can count 12 clusters. What do I miss?

Line 247. Please explain under Table 1 the abbreviations LLR, LSI, and MI. Average year in Table 1 in what is referred to? Please explain.

The discussion and conclusions sections are well-written.

Reviewer 2 Report

Comments and Suggestions for Authors

Review

“Neuropsychological Diagnosis and Assessment of Alexia:

A Mixed-Methods Study”

General assessment:

- This article explores the characterization of diagnostic and assessment-related aspects of alexia in a corpus of literature including recent studies on this condition. The author presents a conceptual model that is supposed to integrate different implications of alexia into a coherent framework. On the whole, the aim of the paper is to serve as a reference point to inform clinical practice and academic research on alexia. The methods basically consist in reviewing a corpus articles retrieved via CiteSpace and VOSviewer.

- The paper could per se be relevant to the scope of the journal. However, the general impression that a theoretically-informed reader gets is that after reading the paper, there is not much information that can be drawn from the author’s analysis. That is, the content could be summarized in less than 18 pages and many redundant parts should be revised/deleted. The core of the paper is represented by the conceptual model presented in Section 4. I am not entirely convinced that the results of the reviewing process can be considered highly relevant for the scientific community as they are treated in this paper, but it has to be acknowledged that the author has done a lot of research to synthesize the content of the studies considered.

- I have the impression that the author has used AI to generate some parts of the text. Please revise the corresponding parts adapting the register, which is not always appropriate in a scientific paper like this.

My overall recommendation is:

ACCEPT AFTER MAJOR REVISION

Content:

- I find that the abstract should be revised and made more readable. In the present form, it is not very informative (e.g. “…which culminated in a conceptual model portraying the interplay between neural, cognitive, and diagnostic dimensions, aligning with the International Classification of Functioning, Disability, and Health (ICFDH) framework.” > How did this culminate in a model? How is “model” supposed to be interpreted in an abstract if the reader still does not know what the author has done?). As someone who works in the realm of AI, I have the feeling that this abstract has been (partly) generated by ChatGPT. If this is the case, please provide a “human” version of it. The abstract should clarify in simple words what you did, how you did it and what the main results are.

- l. 39-41: The depiction of phonological alexia is quite inaccurate. What is “phonological” about this subclass exactly? Does this specific condition result from the incapability of deciphering phoneme-grapheme correspondences?

- What I find is missing in the central and in the final part of the paper (Section 4 and 5) is a more detailed discussion of how the model developed here is to be used exactly by academic research and clinical practice. The discussion of these implications is what should remain after reading the paper. I must confess that especially the section labeled “Discussion” (Section 5) got me a bit confused, since I have the impression that what is discussed here is not really the connection between what is treated in the relevant literature and the model proposed above (which is what I expected), but rather further aspects mentioned in the studies consulted. This part, however, comes after the presentation of the model, and I think the two components should be discussed in combination here.

- l. 401-402: “The discussion of alexia within the broader context of previous literature underscores the disorder's heterogeneity and the need for an equally diverse diagnostic approach.” > I do not find this particularly surprising… How is this piece of information relevant to what is being said here?

- I think Section 6 has a wrong title and contains what is supposed to be in Section 7; the same goes for Section 7.

Formal aspects:

- p. 1: After “* Correspondence:”, something must be missing. Maybe this indication is in the wrong position and should appear above the names?

- p. 1: “Analysing 449 articles, with CiteSpace and VOSviewer software, …” > The analysis of 449 articles with CiteSpace and the VOSviewer …

- p. 1: “manifests in myriad forms” > This seems to be a bit exaggerate. This is a medical condition and the forms in which alexia manifests itself are limited and circumscribable (the attribute “myriad” is not adequate in this register).

- p. 1: exiting > existing?

- l. 54: “an acquired reading disorder” > Why this specification? This has already been pointed out above.

- l. 61: “(Cohen et al., 2016” > Shouldn’t this correspond to a number referring to a specific position in the References?

- l. 66-67: “The neurofunctional landscape of pure alexia is further nuanced by the observation of broader visual processing impairments” > I find that this type of language is not appropriate in a paper like this. What is this supposed to mean? Please go straight to the point and do not use poetic imagery in an article about language impairment.

- l. 78-79: “a complex neurocognitive disorder characterized by impaired reading abilities” > Why are you repeating this?

- l. 206-208: “In the domain of neuropsychological research, the classification of literature into coherent clusters serves as a powerful tool for elucidating the multifaceted nature of alexia and its assessment.” > This is quite redundant.

- l. 403: “Damasio and Damasio (1983)” > Should this correspond to a number referring to a given position in the References?

Comments on the Quality of English Language

(s. above)

Round 2

Reviewer 1 Report

Comments and Suggestions for Authors

Manuscript ID: brainsci-3051453

Title: “Neuropsychological Diagnosis and Assessment of Alexia: A Mixed-Methods Study”

Thank you for the opportunity to review the revised version of the article entitled: “Neuropsychological Diagnosis and Assessment of Alexia: A Mixed-Methods Study”.

I propose some minor changes that will enhance the readability of the manuscript.

Below the authors will find my comments.

Abstract

The abstract is over 200 words. To comply with the Journal’s guidelines, do the following:

Line 13. Delete the phrase “Addressing this challenge” and write article the with a capital letter.

Line 16. Delete the phrase “as our bibliographic databases”

Line 17. Delete the words “matching or”

Line 21. Delete the phrase “like ‘neural substrates of reading’ and ‘visual processing and perceptual deficits’”.

Line 26. Delete the phrase “These findings suggest that” and start the word future with a capital letter.

Introduction

Line 62. Replace the phrase “visual word form area” with the abbreviation you have used two lines above.

Line 67. Do not start a new paragraph. Unify lines 54 to 76 into one paragraph.

Lines 78-79. Delete the phrase “is complemented by neuropsychological assessments that” and replace it with “that among others”

Findings

Line 327. Use the abbreviation for the phrase “visual word form area”

Line 373. Use only the abbreviation, you have presented its meaning above in your manuscript.

Lines 383-384 Write the word “factor” with a lower letter and do the same for the words “environmental” and “factors”

Author Response

Abstract

The abstract is over 200 words. To comply with the Journal’s guidelines, do the following:

Thank you. The abstract is now 200 words exactly. 

Line 13. Delete the phrase “Addressing this challenge” and write article the with a capital letter.

Thank you. Done. 

Line 16. Delete the phrase “as our bibliographic databases”

Thank you. Done. 

Line 17. Delete the words “matching or”

Thank you. Done. 

Line 21. Delete the phrase “like ‘neural substrates of reading’ and ‘visual processing and perceptual deficits’”.

Thank you. Done. 

Line 26. Delete the phrase “These findings suggest that” and start the word future with a capital letter.

 Thank you. Done. 

Introduction

Line 62. Replace the phrase “visual word form area” with the abbreviation you have used two lines above.

 Thank you. Done. 

Line 67. Do not start a new paragraph. Unify lines 54 to 76 into one paragraph.

 Thank you. Done. 

Lines 78-79. Delete the phrase “is complemented by neuropsychological assessments that” and replace it with “that among others”

 Thank you. Done. 

Findings

Line 327. Use the abbreviation for the phrase “visual word form area”

 Thank you. Done. 

Line 373. Use only the abbreviation, you have presented its meaning above in your manuscript.

 Thank you. Done. 

Lines 383-384 Write the word “factor” with a lower letter and do the same for the words “environmental” and “factors”

 Thank you. Done. 

Reviewer 2 Report

Comments and Suggestions for Authors

The paper can be published in present form.

Author Response

Thank you very much for your effort on helping us to revise and improve this paper.